# Furcellaran-Coated Microcapsules as Carriers of *Cyprinus carpio* Skin-Derived Antioxidant Hydrolysate: An In Vitro and In Vivo Study

**DOI:** 10.3390/nu11102502

**Published:** 2019-10-17

**Authors:** Joanna Tkaczewska, Ewelina Jamróz, Ewa Piątkowska, Barbara Borczak, Joanna Kapusta-Duch, Małgorzata Morawska

**Affiliations:** 1Department of Animal Product Technology, Faculty of Food Technology, University of Agriculture in Krakow, Balicka 122 street, 30–149 Krakow, Poland; 2Department of Chemistry, Faculty of Food Technology, University of Agriculture, Balicka 122 Street, 30–149 Krakow, Poland; 3Department of Human Nutrition and Dietetics, Faculty of Food Technology, University of Agriculture in Krakow, Balicka 122 Street, 30–149 Krakow, Poland; 4Department of Sports Medicine and Human Nutrition, Institute of Human Physiology, University of Physical Education in Krakow, Jana Pawla II 78 Street, 31–537 Krakow, Poland

**Keywords:** carp, gelatine, animal study, antioxidant activity, microcapsules, furcellaran

## Abstract

Carp skin gelatine hydrolysate (CSGH) may be a possible bioactive peptide source, as promising antioxidant properties have been noted during in vivo testing. Hence, the present study focused on improving the bioavailability of the antioxidant peptides from CSGH and on the use of furcellaran (FUR), which can protect the biopeptides during digestion in the gastrointestinal tract. Therefore, in this study, microcapsules coated with furcellaran and containing CSGH cores were prepared. The structural properties of the sample were determined using FT-IR and SEM analysis. The antioxidant properties of hydrolysate, uncoated, and encapsulated samples were investigated. In vivo analysis included determination of its safety in an animal organism and evaluation of the lipid profile, antioxidant blood status, and mRNA expression of some genes involved in antioxidant status in Wistar rats. The results showed no adverse effects of microencapsulated protein hydrolysates in laboratory animals. Nonetheless, there was a statistically significant rise in the level of total antioxidant status blood serum among animals consuming CSGH and not inducing oxidative stress. This can be viewed as a promising indication of the positive effects of antioxidant properties tested in vivo. The process of CSGH microencapsulation in FUR cause a decrease in antioxidant hydrolysate activity, both in vitro, as well as in healthy Wistar rats. When considering the results of the presented diverse therapeutic potential, further research on CSGH being a potential bioactive peptide source used as a functional food or nutraceutical, but with a different microencapsulation coating, is encouraged.

## 1. Introduction

Currently, we expect foods to satisfy hunger while delivering nutrients essential for humans, but their task is also the prevention of diet-related diseases. Functional foods play a significant role in this case. Due to their beneficial physiological effects, including antioxidant, anticancer, and anti-inflammatory properties, nutrients from marine animals, as well as bioactive components have great potential as ingredients in functional foods [1].

After filleting, fish processing waste comprises 75% of the overall fish weight. Approximately 30% of the waste is in the form of bones and skins. Gelatine can be derived from the processing of fish skins, solving the problem of waste disposal while producing a value-added product [2]. Its fast growth rate, as well as good fodder usage cause carp (*Cyprinus carpio*) to be one of the most commonly-bred species in eastern Europe, as well as in Poland. The annual European carp production totals approximately 4,328,083 tons [3]. In our previous research, we demonstrated that protein hydrolysates from carp skin gelatine hydrolysate (CSGH) can be potential bioactive peptide sources [3].

Bioactive peptides consisting of 2–20 residues of amino acids are inactive within the sequences of parent proteins. They can be released via enzymatic hydrolysis either at the time of digestion or during food processing. Antioxidant peptides are among the most frequently-studied bioactive peptides. In particular, in the health food and pharmaceutical industry, fish protein hydrolysates with antioxidant properties have become a highly interesting and discussed subject [4].

There can be a reduction in bioactive peptide activity due to susceptibility to proteolytic degradation or undesirable interactions with other compounds. Encapsulation usage is a highly significant factor in the protection of these biomolecules, consequently providing greater therapeutic efficacy. Encapsulation may also permit the development of the delivery system through the improvement of stability, causing an increase in residence time regarding circulation, while at the same time causing a significant decrease in toxicity [5].

The encapsulation of bioactive ingredients within colloidal particles is an influential means of protecting them from degradation. Different types of colloidal delivery systems have been developed, for example emulsions, nanoemulsions, microemulsions, liposomes, solid lipid nanoparticles, hydrogel beads, and biopolymer nanoparticles, each with its own advantages and disadvantages [6]. Edible coating materials are usually composed of polysaccharides and proteins [7]. These materials could be used not only in the form of capsules, but also as films covering the food. Edible films and coatings often containing bioactive agents or food additives like nutrients are gaining relevance as potential tools to reduce the deleterious effects of the incidence of microorganisms or to improve the nutritional value of a product [8]. In the research by Sapper et al. [9], polysaccharide protein films with the addition of thyme essential oil were obtained, which has a strong in vitro fungicidal effect. However, in vivo studies did not confirm the inhibitory effect on the surface of apples. Furthermore, other data from the literature indicate that the properties of biologically-active compounds tested in vitro are not always confirmed in vivo [10].

Bioactive peptides differ from other food bioactive compounds like vitamins or polyphenols. There is slight heterogeneity regarding the chemical species in the protein hydrolysates. Most studies on bioactive peptides focus on finding new bioactivity and protein precursors, while limited attention is paid to their biostability or bioavailability. Presently, there is a gap in the literature illustrating the various aspects of encapsulation concerning food-protein-derived bioactive peptides. Moreover, there is a need for in vivo studies applying animal models to confirm the success of encapsulation in bioactivity retention following the oral consumption of these products [6].

Polysaccharides are generally ideal as delivery agents in microcapsules as they are structurally stable, ample in nature, and inexpensive. Polysaccharides with reactive functional groups (carboxyl (COOH)), hydroxyl (OH), amide (NH_2_), and sulphate groups (SO_4_H)) are good candidates as carrier matrices [6]. Furcellaran (FUR) is structurally connected with the algal polysaccharide κ carrageenan; however, there is a major structural difference. Furcellaran, which has units consisting of a fragment of (1–3) β-D-galactopyranose with a sulphate group at C-4 and (1–4) -3, 6-anhydro-α-D-galactopyranose, is used as a gelling and stabilizing agent in the food industry [11]. In addition, furcellaran can form complexes with various proteins [11,12], which can be used for encapsulation in food and agriculture applications. The interactions between proteins and polysaccharides can lead to the creation of completely new materials. Negatively-charged furcellaran can act as a film-forming matrix in active, biopolymer films and as a carrier material in the encapsulation process [12,13,14].

We were able to obtain complexes with the participation of proteins. This is an excellent matrix when forming biopolymer films. In our previous study, we used this polysaccharide for carp skin gelatine hydrolysate complexation, while it is also a film-forming matrix for rosemary extract active coatings. This is a safe, non-toxic, biodegradable, and biocompatible material that is used for film and capsule production [14,15]. Additionally, because of its chemical structure, furcellaran possesses mucoadhesive properties [16]. The implementation of polymers with these characteristics when creating new drug forms allows for a significant extension of the formulation contact time with the affected site. Simultaneously, it ensures that the drug has a longer effect, while by considering substances with systemic activity, bioavailability can be improved in comparison with that of traditional forms [17].

In our previous research, using various in vitro tests, it was shown that CSGH has high antioxidant properties [3]. However, in vivo studies did not clearly confirm the high antioxidant activity of CSGH in a living organism. The level of glutathione reductase in the blood serum of animals fed CSGH was statistically higher than that in the serum of control animals; however, there were no statistically-significant differences in the remaining oxidative stress indices [18]. Therefore, a research hypothesis stating that the microencapsulation of CSGH can protect it against a low pH in the stomach of experimental animals and, thus, increase the antioxidant effect of the in vivo preparation was posed. Therefore, the aim of the study was to examine the possibility of improving the bioavailability of CSGH in living organisms by microencapsulation in furcellaran. Due to the verification of the hypothesis, in this study, microcapsules coated with furcellaran and containing CSGH cores were prepared, and then, the antioxidant properties of non-coated and encapsulated samples were investigated. The inclusion of encapsulated and non-encapsulated hydrolysates in the diets of healthy Wistar rats was aimed at assessing their safety and in vivo antioxidant activity. 

## 2. Materials and Methods

### 2.1. Preparing the Protein Hydrolysate

A carp processor (Sona, Koziegłowy, Poland) was used to obtain carp (*Cyprinus carpio*) skins. The residual tissue was manually removed, and then, the skins were ground using a MADO MEE 613 grinder (Dornhan, Germany). Gelatine was obtained using the method described by Tkaczewska et al. [19]. Hydrolysis was conducted using a Protamex enzymatic mixture (Novozymes, Bagsværd, Denmark). Subsequently, lyophilized carp skin gelatine (10 g) was dissolved in 150 mL of water and then heated to 50 °C. The pH was adjusted to 7 with 1 M of HCl. The enzymatic mixture addition comprised 2% protein. Hydrolysis was performed for 3 h, the pH being constantly monitored and adjusted (1 M NaOH) for the first 15 min of the process, followed by correction after every subsequent 15 min. The reaction was halted by inactivating the enzyme by heating the hydrolysate to a temperature of 95 °C for 15 min. On ice, the samples were cooled, and they were subjected to centrifugation at 8000× *g* at 10 °C for a 15-minute period.

### 2.2. Preparation of Microcapsules

Powdered furcellaran EastGel Type 7000 (FUR) was obtained from Est-Agar AS (Karla village, Estonia). The chemical content of FUR was as follows: carbohydrates, 79.61%; proteins, 1.18%; and fats, 0.24%. A furcellaran solution (0.5 g/40 mL H_2_O) was prepared, which was then stirred for 30 min in a magnetic stirrer (250 rpm) (MR Hei-Tec, Bionovo) at 40 °C. Subsequently, glycerin (0.25 mL) and 1.5 g CSGH were added. After 10 min, 10 mL of H_2_O were added, and the solution prepared in this manner was stirred for another 10 min. The resulting solution was poured into Petri dishes (9 mm in diameter) and was left to dry under a running fume hood for 2 days. The procedure for obtaining microcapsules is described in Figure 1.

### 2.3. FT-IR Spectra

Fourier transform infrared spectroscopy was determined using the MATTSON 3000 FT-IR spectrophotometer (Madison, Wisconsin, USA) with the following parameters: wave number range 400–4000 cm^−1^ and resolution 4 cm^−1^.

### 2.4. Scanning Electron Microscopy 

Dry microcapsules were visualized using scanning electron microscopy (SEM). Photographs were taken using an JEOL JSM-7500F field emission scanning electron microscope at a voltage of 15 keV. 

### 2.5. Determination of Hydrolysate and Its Microcapsule Antioxidant Activity

#### 2.5.1. The FRAP Method

Determination of the sample reducing potential was conducted in accordance with a slightly modified method described by Khantaphant and Benjakul [20]. The oxidant in the FRAP assay comprised an acetate buffer (pH 3.6), solutions of ferric chloride (20 mM), as well as 2,4,6-tripyridyl-s-triazine (10 mM TPTZ in 40 mM HCl) at 10:1:1 (*v*/*v*/*v*), which was freshly prepared on the day of analysis. Vortexed FRAP solution (2.85 mL) was mixed with 150 µL of hydrolysate or microcapsule solution. The tubes were incubated in the dark at 37 °C for a period of 30 min and were then assessed using the Helios Gamma UV-1601 spectrophotometer (Thermo Fisher Scientific, Waltham, MA, USA), The measurement of absorbance was conducted at 593 nm. The calculation of the results was performed on the basis of the μmol of Trolox equivalent per 1 mg of hydrolysate activity.

#### 2.5.2. DPPH Radical Scavenging

DPPH radical scavenging activity was conducted based on the proposal by Wu et al. [21]. A mixture containing 1.5 mL of hydrolysate or microcapsule solution and 1.5 mL of DPPH (0.15 mM) in 95% ethanol was prepared. The mixture absorbance was measured at 517 nm after 30 min. The scavenging effect was calculated with the use of the following formula:DPPH radical scavenging (%) = [(A_blank_ − A_sample_)/A_blank_] · 100%

### 2.6. Effects of Gastrointestinal Proteases on Gelatine Hydrolysate and Microcapsule Antioxidant Activity

As defined by Teixeira et al. [22], an in vitro digestive model system, including enzymes similar to those found in the human upper gastrointestinal digestive tract, was used. The CSGH or its microcapsule solution’s pH was adjusted to 2.0 with HCl (1 M) with the addition of pepsin (Sigma-Aldrich, Poznan, Poland). The solution’s pH was set to 5.3 with 0.9 M NaHCO_3_ following incubation at 37 °C for 1 h. Pancreatin (Sigma-Aldrich, Poznan, Poland) was added, and the pH was adjusted to 7.5 (1 M NaOH), while the mixture was incubated at 37 °C for 2 h. The inactivation of both enzymes was induced by boiling in water for 10 min after the completion of digestion. The digested peptide and hydrolysate solution underwent centrifugation (11,000× *g*; 15 min) while the supernatant was lyophilized. CSGH and microcapsule antioxidant activity were measured following digestion.

### 2.7. Animals and Dietary Treatments

Male Wistar rats (*n* = 24), aged 4–6 weeks old, were purchased from an animal husbandry in Brwinów, Warsaw, Poland. At the beginning of the experiment, the mean body mass of the rats was 123 g. The experimental procedures were approved by the First Local Ethical Committee on Animal Testing, Jagiellonian University in Kraków. Prior to the experiment, for one week, the rodents were acclimatized with standard laboratory chow.

In the collective cages (8 per group) to which the animals were moved, unlimited drinking water access was given, as well as an experimental diet. In Table 1, comprehensive descriptions of the diets can be found. Animals from the control group were given the control AIN-93G diet; group H was fed the AIN-93G diet comprising 1% (*w*/*w*) of freeze-dried carp gelatin hydrolysate; and group CH was fed the AIN-93G diet containing 1.5% (*w*/*w*) of encapsulated freeze-dried carp gelatine hydrolysate. The experimental diet intake was noted for each day. 

The rats were anaesthetized via inhalation using isoflurane following the 35-day experiment period. The blood for testing was obtained by puncturing the heart and was then collected in plain test tubes. The collection of blood samples was performed to obtain serum via centrifugation (1500× *g*, 10 min). The hearts, kidneys, and livers underwent dissection and were subsequently washed in 0.9% sodium chloride, dried with laboratory tissue paper, and finally weighed. The serum was maintained in a frozen state at −80 °C until the time of analysis.

### 2.8. Crude Lipid Levels in Selected Organs

The fat content was determined using the Soxhlet method with a Soxtec Avanti 2050 Auto Extraction Unit (Tecator Foss, Hillerød, Sweden), as previously reported by Kopeć et al. [23].

### 2.9. Analysis of the Serum and Blood

Serum was analyzed to determine the total cholesterol (TC), low-density lipoprotein (LDL-C), high-density lipoprotein (HDL-C), and triacylglyceride (TAG) concentrations (respectively Catalog Numbers ACCENT-200 CHOL, ACCENT-200 LDL, ACCENT-200 HDL, ACCENT-200 TG, PZ Cormay S.A. Lublin, Poland). Aspartate aminotransferase (ALT), uric acid, creatinine, alanine aminotransferase (AST), urea, and the total protein activity level in the serum were determined using kits (Catalog Numbers ACCENT-200 ALAT, ACCENT-200 UA; ACCENT 200 CREAT, ACCENT 200 UREA, ACCENT 200 TP, PZ Cormay S.A. Lublin, Poland, respectively).

An assay kit was used to measure the malondialdehyde level (MDA) (Catalog Number ALX-850-287, Enzo Life Science, Warsaw, Poland), as well as the total antioxidant status (TAS) (Catalog Number NX2332, Randox, United Kingdom), which was performed according to the manufacturer’s guidelines. Glutathione reductase activity (GRS) was measured in the serum using the kit (Catalog Number GR 2368, Randox Laboratories Ltd. Crumlin, UK). Heme oxygenase-1 activity (HO-1) was measured with a kit (ACCENT-200 BIL TOTAL, PZ Cormay S.A. Lublin, Poland) commercially available for bilirubin concentration determination.

### 2.10. Gene Expression

The isolation of RNA from livers, kidneys, as well as hearts was performed using the Total RNA Mini Plus kit (Catalog Number 036–100, A&A Biotechnology, Gdynia, Poland). The RNA content was determined using a Multiscan Go spectrophotometer (Thermoscientific, Waltham, MA USA). RNA was reverse transcribed using the TranScriba cDNA Synthesis Kit for cDNA synthesis (Catalog Number 4000–100 A&A Biotechnology, Gdynia, Poland). As previously described, cDNA underwent real-time PCR analysis (CFX96 Touch™ Deep Well Real-Time PCR Detection System, Bio Rad, Hercules, CA, USA) in a mixture reaction containing the TaqMan Gene Expression Master mix (Cat. No. 4369016, Applied Biosystems, Foster City, CA, USA) and primers for the following genes: superoxide dismutase, heme oxygenase-1, and glutathione reductase (Invitrogen, Life Technologies, Oslo, Norway) [23]. Expression rates were determined as the difference in the normalized threshold cycle (CT) between the controls and the samples, with adjustment for amplification efficiency being made relative to the level of expression of the *18S* housekeeping gene.

### 2.11. Statistical Analysis

All data were expressed as the mean ± SEM (for animal analysis) or SD (for other analysis).

The significance of the differences between groups was established using one-way analysis of variance (ANOVA) and the Tukey post-hoc test (*p* < 0.05).

The data were subjected to statistical analysis using STATISTICA 10.0 software (Stat Soft, Inc Tulsa, Oklahoma, NA, USA). The significance of the differences between groups was established using one-way analysis of variance (ANOVA) and the Duncan post-hoc test (*p* < 0.05).

## 3. Results and Discussion

### 3.1. FT-IR Analysis

Furcellaran, a negatively-charged polyelectrolyte, was used to encapsulate carp skin gelatine hydrolysate with antioxidant properties. The FUR coatings were intended to protect and cause controlled release of the hydrolysate in rat organisms. FT-IR analysis was used to determine the functional groups of the composites and to confirm the interactions between the microcapsule components in relation to their physical properties. The structural characteristics determined using FTIR analysis are shown in Table 2.

### 3.2. SEM

SEM images of dried FUR/CSGH microcapsules and dried FUR-CSGH film are shown in Figure 2.

In comparison, the surface of an FUR-CSGH film, which is homogeneous and smooth, is shown. In the SEM images of the FUR-CSGH microcapsule, the spherical shape of the microcapsules can be seen. Additionally, it should be noted that after drying, microcapsules with a granular structure were packed into a heterogeneous surface. Moreover, significant structural differences in the dried surface of the film and microcapsules with the same composition, but with different preparation methods were noticeable.

### 3.3. Antioxidant Activity of Hydrolysate and Its Microcapsules

The antioxidant activity of the CSGH measured by the FRAP method was 3.11 μM Trolox/mg sample, lower than that of its microcapsules (4.03 μM Trolox/mg sample) (Table 3). This is in accordance with the findings of Mosquera et al. [24] who reported higher antioxidant capacity (by approximately two-fold) for the peptidic fraction of sea bream scales encapsulated in phosphatidylcholine (PC) nanovesicles. This difference may be attributed to the reducing ability shown by furcellaran. According to Sokolova et al. [25], polysaccharides from algae are active reducing agents regarding ferric ions.

The antioxidant activity of CSGH, evaluated via the DPPH method, was 69.85%, and a significant reduction in radical scavenging activity was observed after encapsulation (Table 3). This is in contrast to the findings of [26], who reported that no significant differences were observed between the antioxidant activity of free and encapsulated protein hydrolysates. These lower values in the DPPH activity of microencapsulates compared with that of the pure hydrolysate could be due to the lower amount of protein hydrolysate in the nucleus of the capsule when compared with that in the pure compound. Given the above result, in animal studies, more microcapsules were added to the diet of the animals from the microcapsule group (1.5%) compared to the group fed pure hydrolysate (1%) so that the antioxidant effect of the hydrolysate would be comparable between groups.

As the next step, CSGH, as well as its microcapsules underwent simulated gastrointestinal digestion using pepsin and pancreatin in order to assess both their stability and antioxidant activity following treatment using gastrointestinal proteases. There were statistically-significant changes in the activity of antioxidants following digestion. The antioxidant activity of the digested samples was evaluated via the FRAP and DPPH methods, and this assessment showed significantly lower values for this activity. Comparable results were achieved by Gómez-Mascaraque et al. [27]. In the latter study, the influence of microencapsulation on protective function at the time of digestion, as well as the stability of the potentially bioactive hydrolysate of whey protein were evaluated. The results suggested that the digestion of these samples altered the hydrolysate peptide profile, causing the number of peptides to be lower, as well as reducing their molecular weights. There was also no protective effect at the time of digestion regarding encapsulation within chitosan microparticles.

In a different study [28], bioactive peptide biostability was evaluated in the case of peptides encapsulated using carboxymethylated gum and sodium alginate. Only minimal (a maximum of 10%) protein material release was noted following gastric simulation. Nevertheless, in the opinion of these authors, the peptides released during the intestinal phase may be present for absorption into enterocytes and consequently into the circulation. At this time, they are still susceptible to additional peptidolytic modification. Hence, it is vital to further assess the antioxidant activity of CSGH, as well as its microcapsules in vivo on the basis of an animal model.

### 3.4. Animal Study

#### 3.4.1. Body Mass Gain, Crude Lipid Content, and Weights of Selected Organs

The body mass gain, kidney, liver, and heart weights were not affected by the various dietary treatments (Table 4). It was suggested that the 1% addition of CSGH or encapsulated CSGH is safe for animals and has no effect on these parameters. 

The crude lipid level in the kidneys and liver was not affected by dietary treatments. A lower content of crude lipids in the heart was measured in groups fed with the encapsulated CSGH diets compared with the hearts of animals fed the control diet. Furcellaran, when used as a CSGH microencapsulation coating, has properties similar to carrageenans and is not assimilated by the human body, providing only fiber with no nutritional value [29]. According to the data from the literature, the addition of soluble dietary fiber fractions to animal diets can cause a lower level of fat accumulation in their organs [30].

#### 3.4.2. Evaluation of Chosen Hepatic and Kidney Functions and Toxic Effects

Aspartate transaminase (AST) and alanine transaminase (ALT) are enzymes that are pathophysiological markers used to assess tissue damage [31]. Moreover, the level of lactate dehydrogenase (LDH) is used to assess hepatic disorders. Active liver damage can be indicated by an increase in the levels of the given markers in blood plasma. CSGH and its microcapsules did not show any such side effects in rat organisms. The LDH, ALT, and AST levels in the serum of animals fed CSGH and its microcapsules were lower than those in the serum of the control animals. However, these differences were not statistically significant (Table 5). These results are in agreement with those of Salem et al. [32], who reported that octopus protein hydrolysates reduced the plasma ALT and AST levels in cholesterol-fed rats.

Creatinine and uric acid are products of metabolic waste, which are removed from the blood in a natural form via the kidneys through glomerular filtration. Therefore, an increase in their levels is totally due to the problem of renal failure. Consequently, toxicity indices of the kidneys, as well as total protein levels in blood serum were additionally tested. The mean values of urea, creatinine, uric acid, and total protein in the rats’ blood did not significantly differ between groups (Table 5). These results are in accordance with the data from the literature, which show that protein hydrolysate not only does not demonstrate any toxic activity on the kidneys, but even indicates a preventive role against kidney deterioration and uric acid hyperfiltration [33,34]. 

Based on the obtained results, it can be assumed that CSGH and its microcapsules are not toxic to the kidneys and livers of laboratory animals. In addition, throughout the experiment, the animals experienced similar weight gains and were in good condition. According to the data from studies conducted to date [35], there has been little concern regarding the safety of food protein-derived hydrolysates, since the body would normally hydrolyze food proteins into peptides and food-grade enzymes anyway, and these processes are utilized for the industrial production of hydrolysates.

#### 3.4.3. Evaluation of Antioxidant Status

The TAS level in the serum of the tested animals differed significantly depending on the applied nutrition. The highest value of the TAS index was observed in the blood serum of animals fed with the addition of CSGH (1.05 μmol/L), while the lowest level of this parameter was found in the blood of animals fed with the addition of microencapsulated CSGH (0.65 μmol/L). These results are consistent with the results of in vitro studies in which the microencapsulated hydrolysate was found to have less antioxidant activity as measured by DPPH than the non-microencapsulated hydrolysate.

Digestion of proteins in the digestive tract begins in the stomach and continues in the lumen of the small intestine. The absorption of free amino acids takes place via active transport. There is evidence that bioactive peptides derived from food proteins can be absorbed and enter the bloodstream unchanged and then affect the physiological functions of the body [36]. It can be assumed that biologically-active peptides with antioxidant properties contained in CSGH retain their stable, original structures during digestion in the gastrointestinal tracts of animals, resulting in an increase in the TAS index in the blood serum of rats. The microencapsulation process of CSGH with furcellaran reduced the antioxidant activity of the hydrolysate in the bodies of animals. Non-starch polysaccharides, which include the furcellaran microcapsule shell, may be partially degraded in the small bowel. Polysaccharides that are not digested pass into the colon unchanged, where they are fermented [37]. It is assumed that furcellaran did not undergo digestion in the small intestine of the tested animals; thus, the hydrolysate was not released from the microcapsules and did not show antioxidant activity. 

The drying process of the microcapsules may hinder the opening of FUR coatings, facilitating the release of the CSGH. Moreover, part of the furcellaran may not complex with CSGH, and in free form, it has been found in the gastrointestinal tracts of rats. According to the data from the literature [38], the by-product of the carrageenan acid hydrolysis process can enolize and autoxidize with concomitant production of the superoxide radical, which may possess strong oxidant activity. This may be the cause of a significantly lower TAS index in the blood of rats fed microencapsulated CSGH compared with animals fed only hydrolysate and the control diet.

MDA, which is a lipid peroxidation end-product, is commonly applied as an oxidative stress marker. In Table 5, MDA levels marked in the experimental rats’ blood are demonstrated.

A significantly lower level of malonaldehyde was observed in the blood of animals fed with CSGH encapsulated in microcapsules compared with animals fed the CSGH diet not subjected to microencapsulation. According to Jamróz et al. [39], furcellaran has naturally-occurring polyphenols with antioxidant activity. It has been proven that polyphenols derived from marine algae have the ability to suppress the increase in lipid peroxidation levels in laboratory animal organisms [40]. It can be assumed that the effect of lowering the level of MDA in the blood serum of rats fed with the addition of microencapsulated CSGH was not due to the action of the hydrolysate, but was because of the presence of polyphenols. However, further research should be carried out to confirm this information.

Enzymes such as catalase, superoxide dismutase, glutathione peroxidase, and glutathione reductase form the enzyme antioxidant barrier. These enzymes interact with one another under physiological conditions, which is the reason why the body’s antioxidant defense is weakened by the inactivation of any of the mentioned enzymes [41]. We determined the effect of CSGH administration, as well as its microcapsules on the antioxidant enzyme activity among rats (glutathione reductase and heme oxidase). 

There was no significant difference with regard to the applied diet regarding the amount of heme oxygenase-1 in the tested animals’ serum (Table 5). Under normal conditions, this enzyme can only be found in the spleen or liver. There, in large quantities, it is involved in aging erythrocyte hemoglobin catabolism. HO-1 gene expression causes an increase in variables when influenced by various factors, including ultraviolet radiation, cell treatment with proinflammatory cytokines, heavy metals, prostaglandins, ethanol, and reactive oxygen species [42]. Moreover, on the basis of the implemented nutrition, there were no significant differences in the activity of glutathione reductase in the blood serum of the animals under study. The healthy laboratory animals used in this study did not experience the induction of oxidative stress, and furthermore, the rats were not exposed to an increase in the production of heme oxygenase-1. Consequently, it may be difficult to observe an increase in the activity of enzymes because of the activity of hydrolysate, as well as its microcapsules. This may be the reason why there were no significant statistical differences between groups. These conclusions agree with those presented in the study by Lassoued et al. [43], who affirmed that experimental animals fed a high-cholesterol diet experience depressed antioxidant systems. After thornback ray hydrolysate was included in the diets of the animals under study, the researchers noted a significant increase in the activity of the antioxidant enzyme in contrast to that in the high-cholesterol group; however, in comparison to the control group, the increase was statistically insignificant. 

There was a statistically significant rise in the level of TAS blood serum among animals consuming CSGH without the induction of oxidative stress. This can be viewed as a promising indication of the positive effects of antioxidant properties tested in vivo. However, further research should be conducted regarding the effects of hydrolysate in animal and human nutrition with induced oxidative stress.

#### 3.4.4. Effect of Tested Diets on Lipid Profile

In terms of diet, modified AIN-93 G had no significant effects. This diet included 1% casein replaced with 1% CSGH. There were no changes induced in TC, LDL-C HDL-C, and TAG levels due to the microcapsule additive. In accordance with various reports in the literature on the subject, fish protein hydrolysate causes a reduction in these parameters’ levels in the blood serum in animals fed diets with high amounts of cholesterol [44,45]. No such effects were observed in healthy Wistar rats, which is in line with the results of our research [46]. In our previous research, there were also no changes in the level of total cholesterol and its HDL and LDL fractions in the blood serum of healthy Wistar rats fed with the addition of 1% or 10% CSGH [18]. According to data from the literature [47], encapsulation leads to successful protection of protein formulations from enzymatic degradation in living organisms, which may increase beneficial effects in living organisms. Microencapsulation of CSGH with furcellaran did not increase its effectiveness in improving the serum lipid profiles of healthy Wistar rats. Therefore, it can be concluded that the lack of the expected effect was not due to the loss of hydrolysate properties in the digestive tract during digestion, but may have been caused by its lack of ability to improve the blood lipid profiles of the experimental animals.

#### 3.4.5. mRNA Gene Expression

In our research, the findings were that mRNA expression regarding the genes under analysis did not undergo changes within the liver, kidneys, or heart (for which data were not presented) in response to varying diet treatments. This may be because the duration of the experiment was short. Therefore, further research should be carried out to determine the impact of the long-term consumption of microencapsulated CSGH on the mRNA expression of genes.

## 4. Conclusions

A way to guarantee that antioxidant treatments reach the site of action is through protection with covers, such as furcellaran. In order to increase the bioavailability and thereby strengthen the antioxidant activity of CSGH in a living body, an attempt was made to design microcapsules with hydrolysate and to assess their in vitro and in vivo effects. In the presented study, biopolymer microcapsules containing CSGH were obtained, the presence of which was confirmed using FT-IR analysis and SEM. On the basis of the achieved results, it appears that the addition of CSGH and its microcapsules in the diets of the subjects for 35 days did not have any effects on the functioning of the liver and kidneys while also being safe for the rodents. The level of TAS in the blood serum increased statistically significantly in the case of healthy animals consuming CSGH supplements. This was achieved without provoking oxidative stress, which is a promising indicator of its antioxidant properties tested in vivo.

The microencapsulation process of CSGH in furcellaran reduces the antioxidant activity of the hydrolysate, both in vitro and in healthy Wistar rats. It can be assumed that furcellaran is not digested in the small intestine of the tested animals, so the hydrolysate was not released from the microcapsules and did not show antioxidant activity. Difficulty with opening microcapsules in the gastrointestinal tract of animals may be the result of drying the microcapsule solution at the production stage. Therefore, analogous analyses should be made using microcapsules in the form of a solution. Further research is also needed regarding the use of other coatings as CSGH microencapsulation matrixes that can more effectively protect the hydrolysate from digestive enzymes, and a form of controlled release of CSGH from biopolymer microcapsules should be developed.

## Figures and Tables

**Figure 1 nutrients-11-02502-f001:**
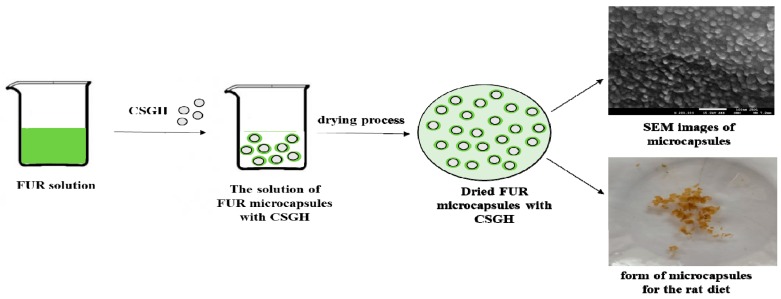
Scheme of furcellaran (FUR) microcapsules with carp skin gelatine hydrolysate (CSGH) formation.

**Figure 2 nutrients-11-02502-f002:**
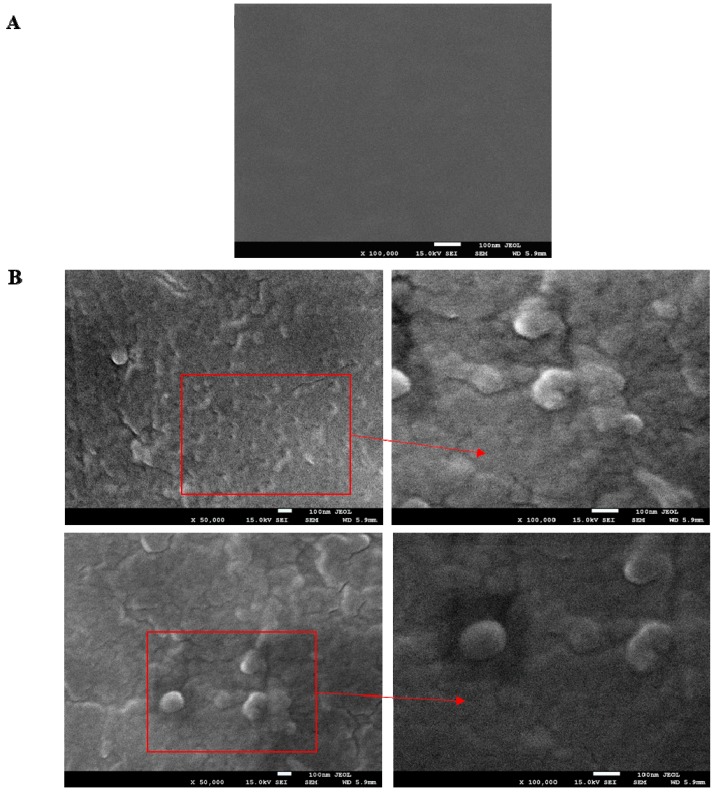
SEM images of (**A**) FUR-CSGH films and (**B**) FUR microcapsules with CSGH.

**Table 1 nutrients-11-02502-t001:** Compositions of experimental diets (g/kg).

Ingredients	Control	H	CH
Corn starch	397.486	387.486	383.196
Maltodextrin	132.00	132.00	132.00
Saccharose	100.00	100.00	100.0
Casein (>85% protein)	200.00	200.00	200.00
Soybean oil	72.00	72.00	72.00
Fiber	50.00	50.00	50.00
Mineral mix ^1^	35.00	35.00	35.00
Vitamin mix ^1^	10.00	10.00	10.00
Choline chloride	2.50	2.50	2.50
TBHQ **	0.014	0.014	0.014
Carp hydrolysate	-	10.00	-
Microencapsulated carp hydrolysate	-	-	14.29

^1^ According to AIN-93G. ** *tert*-butylhydroquinone. Control: group fed the AIN-93G diet. H: group fed the AIN-93G diet with the addition of 1% (*w*/*w*) carp hydrolysate. CH: group fed the AIN-93G diet with the addition of 1.5% (*w*/*w*) encapsulated carp hydrolysate.

**Table 2 nutrients-11-02502-t002:** Characteristic bands of the FT-IR spectra of microcapsules and their components.

Wave Number (cm^−1^) and Band Assignment		
FUR	CSGH	FUR/CSGH Microcapsules
1265 (ester sulphate groups)	-	1260
1064 cm^−1^ (δ_C-O_)	-	1052
926 cm^−1^ (stretching mode of the SO groups)	-	924
-	1647 (amide-I, CO and CN stretching)	1660
-	1534 (amide-II)	1520

The FT-IR spectra of microcapsules can be observed in the characteristic peaks coming from both furcellaran and gelatin hydrolysate. It can be concluded that there is an interaction between negatively-charged furcellaran and positively-charged CSGH groups. In our previous research, this compatibility between compounds was used to obtain furcellaran-gelatine hydrolysate film, while in this study, biopolymer microcapsules were obtained [15].

**Table 3 nutrients-11-02502-t003:** Antioxidant properties of the CSGH and its microcapsules and their stability after digestion.

	Before in Vitro Digestion		After in Vitro Digestion	
	FRAP (μM Trolox/mg sample)	DPPH scavenging (%)	FRAP (μM Trolox/mg sample)	DPPH scavenging (%)
Hydrolysate	3.11 ^bA^ ± 0.02	69.85 ^bB^ ± 0.74	2.85 ^aB^ ± 0.14	41.17 ^aA^ ± 1.95
Microencapsulated hydrolysate	4.03 ^bB^ ± 0.90	43.67 ^bA^± 1.02	2.36 ^aA^ ± 0.06	27.25 ^aA^ ± 3.13

Data are the mean ± SD values. Values marked by different letters are significantly different, *p* ≤ 0.05; a,b: within the same row, separate for the Frap and DPPH assays, A,B: within the same column, separate for the Frap and DPPH assays.

**Table 4 nutrients-11-02502-t004:** Body mass and weights of selected organs (g/kg b.m.).

Treatment	Control	H	CH
Body mass gain and weights of selected organs of experimental rats (g)			
Body gain	334.00 ± 17.2 ^a^	338.50 ± 10.40 ^a^	360.25 ± 14.60 ^a^
Liver	41.79 ± 0.81 ^a^	39.34 ± 0.89 ^a^	41.66 ± 1.16 ^a^
Heart	2.92 ± 0.10 ^a^	3.00 ± 0.10 ^a^	3.14 ± 0.11 ^a^
Kidneys	6.63 ± 0.17 ^a^	6.40 ± 0.19 ^a^	6.63 ± 0.13 ^a^
Crude lipid content in selected organs (% per b.m.)			
Liver	13.88 ± 1.39 ^a^	12.80 ± 1.4 ^a^	10.91 ± 0.95 ^a^
Heart	16.24 ± 4.36 ^b^	14.79 ± 0.59 ^ab^	13.61 ± 0.33 ^a^
Kidneys	6.82 ± 0.85 ^a^	6.03 ± 0.40 ^a^	7.24 ± 0.59 ^a^

Data are mean ± SEM values. Values in rows marked by different letters (a, b, c) are significantly different, *p* ≤ 0.05.

**Table 5 nutrients-11-02502-t005:** Blood parameters of animals.

Treatment	Control	H	CH
TC (mmol/L)	1.89 ± 0.26 ^a^	1.76 ± 0.23 ^a^	1.78 ± 0.11 ^a^
LDL-C (mmol/L)	0.46 ± 0.49 ^a^	0.41 ± 0.29 ^a^	0.37 ± 0.21 ^a^
HDL-C (mmol/L)	0.83 ± 0.60 ^a^	0.68 ± 0.32 ^a^	0.77 ± 0.16 ^a^
TAG (mmol/L)	2.66 ± 0.46 ^a^	2.50 ± 0.09 ^a^	3.16 ± 0.54 ^a^
AST (u/L)	97.6 ± 7.38 ^a^	90.23 ± 3.03 ^a^	92.95 ± 4.87 ^a^
ALT (u/L)	46.83 ± 3.63 ^a^	39.31 ± 3.00 ^a^	43.21 ± 3.4 ^a^
Uric acid (µmol/L)	67.63 ± 6.59 ^a^	59.25 ± 4.31 ^a^	61.38 ± 5.58 ^a^
Creatinine (mmol/L)	22.38 ± 1.83 ^a^	21.25 ± 1.03 ^a^	19.13 ± 0.61 ^a^
Urea (mmol/L)	9.16 ± 1.17 ^a^	7.45 ± 0.38 ^a^	9.36 ± 0.40 ^a^
LDH (u/L)	399.13 ± 65.38 ^a^	293.63 ± 29.62 ^a^	305.25 ± 22.36 ^a^
Total serum protein (g/L)	67.00 ± 4.52 ^a^	63.46 ± 1.11 ^a^	65.68 ± 0.89 ^a^
MDA (nmol/mL)	22.44 ± 1.48 ^ab^	23.88 ± 1.43 ^b^	19.98 ± 0.69 ^a^
TAS (µmol/L)	0.92 ± 0.03 ^b^	1.05 ± 0.03 ^a^	0.65 ± 0.02 ^c^
GSR (U/L)	20.24 ± 1.83 ^a^	17.65 ± 3.19 ^a^	23.67 ± 2.86 ^a^
HO-1 (mmol/L)	2.44 ± 0.06 ^a^	2.56 ± 0.08 ^a^	2.48 ± 0.11 ^a^

Data are the mean ± SEM values. Values in rows marked by different letters (a, b, c) are significantly different, *p* ≤ 0.05. The control group was fed the AIN-93G diet. The H group was fed the AIN-93G diet with the addition of 1% carp hydrolysate. The CH group was fed the AIN-93G diet with the addition of 1.5% encapsulated carp hydrolysate.

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
