# Peer review of "Furcellaran-Coated Microcapsules as Carriers of Cyprinus carpio Skin-Derived Antioxidant Hydrolysate: An In Vitro and In Vivo Study"

_nutrients, 2019, doi:10.3390/nu11102502_

Round 1

Reviewer 1 Report

The article deals with the in vitro and in vivo physiological effects (on healthy Wistar rats) of Cyprinus carpio skin derived antioxidant hydrolysate (CSGH), both as such and encapsulated with furcellaran, an algae-derived polysaccharide. Special focus was on serum lipids and antioxidant status.

Basically, negative results arose from the study, since the encapsulation led to a reduction of the antioxidant activity, and little or no side effects.

First, I found the grammar and language seriously insufficient, so much to substantially affect the readability of the manuscript. Thus, an extensive check is needed and strongly recommended.

Since I have tried to fix most (but not all) of the language mistakes, as well as suggested some rephrasing, I have produced many comments, that the Authors could better track on the herein enclosed document (the manuscript, annotated with my comments).

In the same enclosed documents, the Authors will find also more substantial comments.

For example, in the Introduction, also in the light of the substantially "negative" results arising from this study, more information about coating materials for food/nutrients/additives should be supplied. Such as based on recent articles, for example:
1. Puscaselu, Biopolymer-based Films Enriched with Stevia rebaudiana Used for the Development of Edible and Soluble Packaging. Coatings 2019, 9, 360, doi:10.3390/coatings9060360.
2. Sapper et al., Antifungal Starch–Gellan Edible Coatings with Thyme Essential Oil for the Postharvest Preservation of Apple and Persimmon. Coatings 2019, 9, 333, doi:10.3390/coatings9050333.

Moreover, the statements in lines 71-74 must be explained, supported and interconnected. The reader cannot understand what is the link between these information.

The Authors are invited to check all my comments in the herein enclosed document, respond and react accordingly, and carefully revise the grammar and language, before the manuscript could be considered for publication.

Reviewer 2 Report

The paper by Tkaczewska, J. is interesting but I am missing the major points of the article. I think the writing arrangement of the article has to be changed. What is the hypothesis of the paper and how the authors have done the hypothesis-driven research? Also, a clear description of the questions in the introduction and conclusion will improve the manuscript.

Round 2

Reviewer 1 Report

After reading carefully the revised version of the Manuscript, I can safely state that it has achieved a full academic status. This revision accounts for all my comments and the important comments from the other esteemed Reviewer.

Now, the manuscript can be published, as it is well documented, quite informative, attractive for readers, as well as can lead to further research.